# Analysis of the Physicochemical Properties, Replication and Pathophysiology of a Massively Glycosylated Hepatitis B Virus HBsAg Escape Mutant

**DOI:** 10.3390/v13112328

**Published:** 2021-11-22

**Authors:** Md. Golzar Hossain, Yadarat Suwanmanee, Kaili Du, Keiji Ueda

**Affiliations:** 1Department of Microbiology and Hygiene, Bangladesh Agricultural University, Mymensingh 2202, Bangladesh; mghossain@bau.edu.bd; 2Department of Microbiology and Immunology, Division of Virology, Osaka University Graduate School of Medicine, Osaka 565-0871, Japan; ysuwanmanee@virus.med.osaka-u.ac.jp (Y.S.); dukaili681@virus.med.osaka-u.ac.jp (K.D.)

**Keywords:** hepatitis B virus, replication, HBsAg diagnostic escape mutants, glycosylation, egress

## Abstract

Mutations in HBsAg, the surface antigen of the hepatitis B virus (HBV), might affect the serum HBV DNA level of HBV-infected patients, since the reverse transcriptase (RT) domain of HBV polymerase overlaps with the HBsAg-coding region. We previously identified a diagnostic escape mutant (W3S) HBV that produces massively glycosylated HBsAg. In this study, we constructed an HBV-producing vector that expresses W3S HBs (pHB-W3S) along with a wild-type HBV-producing plasmid (pHB-WT) in order to analyze the physicochemical properties, replication, and antiviral drug response of the mutant. Transfection of either pHB-WT or W3S into HepG2 cells yielded similar CsCl density profiles and eAg expression, as did transfection of a glycosylation defective mutant, pHB-W3S (N146G), in which a glycosylation site at the 146aa asparagine (N) site of HBs was mutated to glycine (G). Virion secretion, however, seemed to be severely impaired in cases of pHB-W3S and pHB-W3S (N146G), compared with pHB-WT, as determined by qPCR and Southern blot analysis. Furthermore, inhibition of glycosylation using tunicamycin^TM^ on wild-type HBV production also reduced the virion secretion. These results suggested that the HBV core and Dane particle could be formed either by massively glycosylated or glycosylation-defective HBsAg, but reduced and/or almost completely blocked the virion secretion efficiency, indicating that balanced glycosylation of HBsAg is required for efficient release of HBV, and mutations inducing an imbalanced glycosylation of HBs would cause the virion to become stuck in the cells, which might be associated with various pathogeneses due to HBV infection.

## 1. Introduction

Hepatitis B virus (HBV) infection is a serious global health problem, with two hundred million people currently infected worldwide. HBV infection leads to both acute and chronic liver disease, followed by liver cirrhosis and hepatocellular carcinomas (HCC) [1]. HBV is one of the smallest enveloped animal viruses, and consists of an icosahedral nucleocapsid surrounded by three kinds of envelope or surface proteins [2]. The viral genome within the nucleocapsid is composed of partially double-stranded circular DNA and encodes four known overlapping open reading frames (ORFs): C (core), P (DNA polymerase, HBV pol), S (surface or envelope proteins: large S [LS], middle S [MS], and small S [SS] or HBs) and X (HBx) [3]. Therefore, a mutation in a particular gene, and especially in polymerase, often has a significant effect on the other ORFs overlapping the gene.

The complete infectious HBV virion, also known as Dane particle, enters into hepatocytes through NTCP bound with the preS1 region [4]. Then, the uncoated nucleocapsids are transported to the nucleus, where the covalently closed circular DNA (cccDNA) is formed from relaxed circular DNA (rcDNA), which is the viral genome in the infectious particle. The cccDNA is competent for transcription of 3.5 kb pregenomic RNA (pgRNA) and other subgenomic RNAs, such as 2.4 kb, 2.1 and 0.7 kb mRNA [5]. The pgRNA is encapsidated together with HBV pol and is reversely transcribed by pol into minus (−) strand DNA; finally, rcDNA is formed along with (+) strand DNA synthesis. The newly formed nucleocapsid occasionally reenters the nucleus (internal cycle) and otherwise is enveloped to be released (egress) through the ER-Golgi pathway [6,7].

HBV pol is a large, complex and multifunctional protein consisting of four domains—the terminal protein (TP), spacer (SP), reverse transcriptase (RT) and RNase H (RNH) domains—and regulates HBV replication by its enzymatic activities but lacks a proofreading capacity, which leads to a higher mutation rate during the replication process [8,9]. Mutations in the RT domain cause the appearance of mutations in the small HBsAg ORF, due to the fact that ORF S completely overlaps the RT-encoding region [10]. On the other hand, the HBV pol RT domain is the target of antiviral drugs belonging to nucleoside/tide analogues [11]. For example, lamivudine (2′,3′ dideoxy-3′-thiacytidine, 3TC) and emtricitabine (2′,3′-dideoxy-5-fluoro-3′-thiacytidine, FTC) inhibit the viral reverse transcriptase activity; adefovir (9-[2-[bis(pivaloyloxymethoxy)phosphorylmethoxy]ethyl]adenine, ADV) and entecavir (9-[(1S,3R,4S)-4-hydroxy-3-(hydroxymethyl)-2-methylenecyclopentyl]guanine monohydrate, ETV) inhibit reverse transcription of the negative strand DNA synthesis from pgRNA, and synthesis of positive-strand HBV DNA [12,13]. Mutations occurred in the RT domain, while treatment with nucleoside/tide analogues frequently leads to emerging drug-resistant HBV generation, although viral replication is usually seriously affected, decreased or restored by the mutations [9,12].

The HBs region is the main component of HBV envelope proteins and one of the important clinical markers for diagnosis. It is produced in high quantities compared to other viral proteins and is essential for both virion formation and secretion [14]. Recombinant HBsAg produced in yeast has been used as a vaccine to prevent HBV infection [15,16,17,18]. The HBs protein consists of 226 amino acids (aa) with a major hydrophilic region (MHR) ranging from residues 99aa to 169aa [19], which includes a highly antigenic determinant called the “a” determinant (124 aa to 147 aa) [20,21,22].

Due to the presence of an N-linked (N146) glycosylation site at aa 146, HBs is post-translationally modified and produced both as glycosylated (gp27) and non-glycosylated (p24) HBsAg in a nearly 1:1 ratio [19,23]. Although the lack of N146-linked glycosylation does not affect the expression or stability of HBs, it affects HBV virion release detrimentally [23]. Indeed, virion release into medium was found to be almost completely blocked by removal of the N146-linked glycosylation site [24], suggesting that HBV virion release requires at least one N-linked glycosylation site in the HBsAg. On the other hand, it was possible to arrange extra-N-linked glycosylation acceptor sites in the HBs other than the acceptor site at position N146 by site-directed mutagenesis. As a result, HBs was hyperglycosylated compared with the wild type. Interestingly, the hyperglycosylated subviral particles were more efficiently secreted, but the virion assembly, secretion and infectivity were impaired [23]. Additional glycosylation sites on HBsAg other than the acceptor site at position N146, such as T123N or K160N, impaired or reduced the antibody recognition, respectively, while also reducing the virion formation [25]. Moreover, a recent study showed that such additional N-glycosylation mutations were significantly higher in the patient with HCC [26], although the effect and the mechanism of non-glycosylated and hyperglycosylated HBsAg on HCC development remained to be clarified.

In our previous study, using commercial ELISA kits, we found several mutations in the HBsAg coding sequence which led to diagnostic failure [27]. The diagnostic escape HBsAg appeared to be massively glycosylated, with only one N-linked glycosylation site (N146) being conserved, though the massive glycosylation was not completely responsible for the diagnostic escape in our study.

Since several novel mutations in the RT domain were found in our diagnostic escape HBs that had not been previously reported, we investigated the replication, assembly and virion secretion of the diagnostic escape RT/HBs mutant HBV in this report. The results showed that several HBV pol RT domain mutations found due to the HBs escape HBV did not correspond to drug-resistant mutations, as reported previously [28]. Massive glycosylation of HBsAg due to HBs/RT overlapping mutations at the single conserved N-linked glycosylation site did not affect the expression and secretion of the HBs subviral particles and was assembled in Dane particles whose secretion was drastically reduced in contrast.

## 2. Materials and Methods

### 2.1. Plasmid Construction and Cells

An HBV-producing plasmid, pHB-WT, was generated by inserting a 1.3-fold length of the HBV genome (genotype C and subtype adr4 [X01587.1]) at the BamHI site of pBR322. The XbaI-SpeI fragments of the pHB-WT were replaced with the region corresponding to W3S and W3S N146G [27], yielding pHB-W3S and pHB-W3S (N146G).

Cells of a human hepatocellular carcinoma cell line, HepG2 (ATCC^®^ HB-8065^TM^), were maintained in Dulbecco’s Modified Eagle’s Medium (DMEM) with low glucose (1.0 g/L) (Nacalai Tesque, Japan), supplemented with heat inactivated 10% FBS (fetal bovine serum; Equitech-Bio), 100 U/mL penicillin G, 100 µg/mL streptomycin and 0.25 µg/mL amphotericin B (Nacalai Tesque), and the cells were cultured at 37 °C in a 5% CO_2_ atmosphere.

### 2.2. Transfection

Transient transfection into HepG2 cells was performed using GenJet™ in Vitro DNA transfection reagent (Ver. II) (SignaGen^®^; Catalog #: SL100489) according to the manufacturer’s protocol. For HBV replication analysis, the HepG2 cells were seeded (2 × 10^5^ cells/well) on a 24-well collagen-coated plate (Iwaki) around 24 h before transfection. The cells were then transfected with 1 µg of pHB-WT, pHB-W3S or pHB-W3S (N146G). To evaluate the transfection efficiency, the cells were co-transfected with a β-galactosidase-expressing plasmid (pCMVß [Clontech], 0.1 µg). For immunofluorescence assay (IFA), cells were seeded (1 × 10^5^ cells/well) on an 8-well collagen-coated chamber glass slide (Matsunami Glass™, Japan) and incubated overnight. Cells were then transfected in a similar manner with 1 µg of the respective plasmids. The cell medium was replaced with fresh medium ~5–6 h after transfection and incubated an additional 72 h. For wild-type and mutant virus production, cells (5–6 × 10^6^/dish) were seeded on a 100 mm dish (Iwaki Cytec™) one day before transfection. Transfection was performed with 10 μg plasmid DNA in a similar way, and the medium was refreshed the next day. The cells and the supernatant were harvested either 3 or 5, and 7 days after transfection for ELISA, Western blot, and HBV replication analysis. The virus for CsCl density gradient centrifugation analysis was collected 7 days post-transfection.

For a beta (β) galactosidase assay, the transfected cells were lysed with 100 µL of Glo™ Lysis Buffer (Cat# E266A, Promega). A 10 µL aliquot of the cell lysates was poured into one well of a 96-well flat-bottom plate, and then 190 µL of 1 × Z buffer (0.1 M NaPO_4_^2−^ pH 7.5, 10 mM KCl, 1 mM MgSO_4_, 50 mM 2-mercaptoethanol) containing 12.5 g/mL CPRG (chlorophenol red-β-d-galactopyranoside [Sigma]) was added. The solution was then mixed with shaking for a while and then incubated at 37 °C for 30 min. The reaction was then stopped by addition of 1M Na_2_CO_3_, and the optical density (OD) was measured at 574 nm with a Spectra Max™ 190 microplate spectrophotometer (Molecular Devices).

### 2.3. Tunicamycin^TM^ Treatment and Cell Viability Assay

The medium of HepG2 cells transfected as described above was replaced with fresh medium containing tunicamycin at the indicated concentration (0, 0.5 or 1.0 µM) at 5–6 h after transfection to inhibit N-linked glycosylation. The corresponding amount of dimethyl sulfoxide (DMSO) was added to an identical medium containing transfected HepG2 cells as a control, since the tunicamycin™ (Sigma) was dissolved in DMSO. Three days later, the medium was again replaced with a fresh medium containing tunicamycin™. The cells were further incubated for 3 days (for a total of 6 days post-transfection/treatment), and then the medium and cells were harvested for further analysis.

To assess the viability of tunicamycin™-treated cells, HepG2 cells were seeded on a collagen-coated 96-well (3 × 10^4^ cells/well) clear-bottom white plate (Corning) with 100 µL culture medium and incubated for 24 h. Then, the cells were treated with the indicated concentration of tunicamycin™ as described above. After three days of incubation, a luminescent cell viability assay solution (CellTiterGlo^®^, Promega, Madison, WI, USA) (100 µL) was mixed into the culture medium and incubated at 37 °C for 15 min. The luminescence was measured using GloMax^®^ GM3000 (Promega) according to the manufacturer’s guidelines.

### 2.4. Virus Preparation and Cesium Chloride (CsCl) Density Gradient Centrifugation

A 5 mL aliquot of supernatant collected from the transfected HepG2 cells was mixed well with 1 mL of 30% (*w*/*v*) PEG8000 (Wako Pure Chemical Industries) solution at a final concentration of 6% and stood still at 4 °C overnight. The mixture was then centrifuged at 8400× *g* for 30 min at 4 °C, and the PEG-precipitated pellet was suspended in 500 μL of TNE solution (10 mM Tris-HCl pH 7.6, 100 mM NaCl, 1 mM EDTA), and the insoluble material was removed by spin down. In the case of pHB-W3S, four 5 mL aliquots (i.e., 20 mL) were used and they were finally combined into one 500 µL TNE solution. The solution was treated with 20 units/mL DNase I and 2.5 μg/mL RNase A for 1 h at 37 °C in the presence of 5 mM MgCl_2_. The DNase I was inactivated by adding a 1/20th volume of 0.2M EDTA pH 8.0 and 0.2M EGTA pH 8.0 and further by incubating for 30 min at 70 °C. Then, the sample was loaded on top of a CsCl gradient solution (39F 350 μL, 35F 250 μL, 31F 250 μL, 27F 250 μL, 23F 250 μL, 19F 250 μL and 15F 150 μL from the bottom, F; CsCl percentage in weight). The sample was centrifuged at 50,000 rpm at 15 °C for 20 h using a Beckman Ti55 rotor and Optima^®^ TLX ultracentrifuge (Beckman) [29]. After centrifugation, twelve fractions (150 µL/each) were collected from the top. A 25 μL aliquot of each fraction was used to measure the refraction value to calculate the density, and 20 μL aliquots of each fraction were used for PreS1 ELISA, DNA extraction, and Western blot analysis, respectively.

### 2.5. Enzyme-Linked Immunosorbent Assay (ELISA)

HBeAg ELISA (HBeAg ELISA kit; Bioneovan, Beijing, China) and preS1 ELISA (Anti-HB Pre-S1 Antigen Quantitative ELISA Kit, Beacle, Kyoto, Japan) were also conducted according to the manufacturer’s instructions, since a commercial HBsAg ELISA could not detect W3S sAg [27]. In these assays, OD_450–630_ was measured by using a Spectra Max 190 microplate spectrophotometer (Molecular Devices).

### 2.6. Western Blotting

The transfected cells were washed with PBS (phosphate buffered saline; 137 mM NaCl, 2.7 mM KCl, 10 mM Na_2_HPO_4_ and KH_2_PO_4_) and lysed in a lysis buffer (50 mM NaH_2_PO_4_ pH 8.0, 300 mM, NaCl, 0.1% NP40 and a complete mammalian protease inhibitor (Sigma P8849; 1:1000 dilution)) to extract total protein. The extracted proteins were used for Western blotting as described previously [27,30]. Western blotting was performed using an antibody targeting the preS1 region (anti-preS1 mono 1; Beacle) [27,30].

### 2.7. Immunofluorescence Analysis (IFA)

Immunofluorescence analysis was performed according to the previously described protocol. Briefly, at around 72 h post-transfection, the transfected cells were washed with PBS. The cells were then fixed with 4% paraformaldehyde for 1 h, washed with PBS, and permeabilized with 0.1% Triton^®^ X-100 in PBS for 30 min. The cells were stained with anti-preS1 antibody (anti-preS1 mono 1; Beacle), followed by a fluorescent-labeled secondary antibody (Alexa Fluor 488, a goat anti-mouse IgG; Molecular Probes). The cell nuclei were then counterstained with DAPI (4′,6-diamidino-2-phenylindole) and the slides were mounted with glycerol for confocal microscopy analysis (TCS SP8; Leica Microsystems).

### 2.8. DNA Extraction and Quantitative Real-Time PCR (qPCR)

The extra-virion DNA and RNA in the supernatant were degraded as described above. The DNA from virus particles was extracted and purified as described elsewhere [31]. For intracellular core DNA extraction, the cells were washed with PBS and then lysed using 500 µL of a hypotonic buffer (20 mM Tris-HCl pH 7.6, 50 mM NaCl, 5 mM MgCl_2_, 1 mM EDTA, 0.1% 2-mercaptoethanol and 1% Nonidet P-40) [32]. The cell debris was excluded by centrifugation for 15 min at 17,000× *g*, and the supernatant containing the HBV core particles was collected. The non-core particle-associated cytoplasmic DNA and RNA were degraded by treating with 2 μL (10 unit) DNase I (Takara-Clontech) and 5 μL (0.5 μg/mL) RNase A (Roche, Switzerland) in the presence of 5 mM MgCl_2_ and 5 mM CaCl_2_ for at least 3 h at 37 °C. The reaction was then stopped by adding EDTA and EGTA at 10 mM, respectively. The sample was disrupted by adding SDS (1% as final concentration) and 0.2 mg/mL proteinase K (Roche, Switzerland) and incubated at 56 °C for at least 3 h. After adding 10 µg (1 µL) of sonicated salmon sperm DNA, the aqueous phase of the sample was extracted with phenol-chloroform-isoamyl alcohol (24:24:1) followed by ethanol precipitation and rinsing with 70% ethanol and drying, and finally, the core-associated HBV DNA was dissolved in 20 µL of TE buffer (10 mM Tris-HCl and 1 mM EDTA pH:8.0). The HBV DNA (1 µL out of 20 µL) was quantified by qPCR with a specific primer set at the HBs region (HBs F2: 5′-CTTCATCCTGCTGCTATGCCT-3′; HBs R2: 5′-AAAGCCCAGGATGATGGGAT-3′). The quantification was carried out using SYBR^®^ Green Master Mix (Thermo Fisher Scientific, Waltham, MA, USA) and a QuantStudio™ 6 Flex real-time PCR system (Thermo Fisher Scientific) following the manufacturer’s guidelines. Triplicate reactions for each sample were performed and the average values were calculated according to the standard line [31].

### 2.9. Southern Blot Analysis of Viral DNA

The full-length HBV DNA fragment (~3.2 kb) was prepared from pBR-3HBneo [33] with BamHI digestion followed by gel-purification and was used as a probe to detect HBV DNA on the Southern blot. The DNA was denatured at 100 °C for 5 min and labeled using Amersham^TM^ AlkPhos Direct Labelling Reagents according to the manufacturer’s protocols (Cat#RPN3680; GE Healthcare).

The extracted intracellular core-associated and extracellular virion-associated and HBV DNA from the transfected cells and medium, respectively, were run on 1.2% TAE agarose gel. After the electrophoresis, the gel was treated with 0.2N HCl for 10 min, 0.5N NaOH-0.6M NaCl for 20 min and 0.5M Tris-HCl (pH 8.0)-1.5M NaCl for 10 min. The DNA in gel was transferred onto a nylon membrane (Amersham Hybond^TM^-N^+^; GE Healthcare) through 0.4N NaOH overnight. The membrane was then cross-linked with UV and washed with 3 x SSC (saline sodium citrate; 20 x SSC, 3 M NaCl-0.3 M 3Na-citrate, pH 7.0) and hybridized with the probe at 55 °C overnight. Then, the membrane was washed with a washing buffer (urea 12%, SDS 0.1%, 50 mM NaHP_2_, 150 mM NaCl, 1.0 mM MgCl_2_ and 0.002% blocking reagent), washed with a detection buffer (50 mM NaCl, 2 mM MgCl_2_ in DWB [1.0 M Tris, pH 10]) and rinsed with CDP-STAR (Cat# 12041677001, Roche) for 5 min for chemiluminescence, which was detected and visualized by a ChemiDoc™ imaging system (Bio-Rad).

### 2.10. Statistical Analyses

Statistical significance was analyzed using a Student’s *t*-test, and values of *p* < 0.05 were considered to indicate statistical significance. Every experiment was performed at least three times independently, and the mean values ± standard deviation (SD) were determined.

## 3. Results

### 3.1. Comparison of HBs and Pol Amino Acids Sequences of HBV-WT (adr4) and W3S

Several amino acid mutations in the RT region of the HBVpol were found in the W3S, since the HBs sequence was overlapped with RT (Figure 1A,B, and the bottom). We reported a diagnostic escape HBs sequence (W3S) in which two mutations, P120K and T123D close to the “a” determinant, were responsible for the escape [27]. In addition, the diagnostic escape HBsAg was massively glycosylated at the conserved N-linked glycosylation site (N146). These mutations had not been reported previously, and so it was an interesting question whether they would have any effect on HBV replication. To examine this, we constructed an HBV replication-competent plasmid pHB-W3S, which was thought to be massively glycosylated on the HBs with several RT mutations (Figure 1B), and the plasmid pHB-W3S (N146G), which was glycosylation-deficient with the same RT mutations as pHB-W3S on the backbone of pHB-WT (adr4, X01587.1) with exception at K152R (Figure 1B) [33,34]. The pol RT K152R mutation was due to the construction of glycosylation-deficient HBsAg N146G (Figure 1B). These plasmids were transiently transfected to HepG2 cells, and three days after transfection, the expressions of HBeAg and preS1 (large S; LS) were determined by ELISA. As shown in Figure 1C, the HBeAg and preS1 (LS) expressions in the cells and HBeAg from the medium were not very different among the wild type (WT) and mutants (W3S, and W3S [N146G]), whereas the HBsAg expression from the mutants could not be detected by ELISA (Figure 1C) as reported [27]. The large S (LS) protein in the lysate was monitored with Western blot analysis, since we had no reliable anti-S antibody in hand and W3S HBsAg was unlikely to be detected. Figure 1D shows the almost comparable expression of LS and the massive glycosylation of W3S LS. The large S from W3S (N146G) canceled the glycosylation (Figure 1D). These results suggested that the W3S HBsAg should be massively glycosylated at N146 even under replication condition, which was consistent with our previous report [27].

### 3.2. Analysis of the Replication Status of HBs/RT Mutant HBV

Next, we tested the replication and virion production efficiency of the W3S mutant HBV compared to the WT. The transfection efficiencies were monitored by β-galactosidase assay (Figure 2A). There was no difference in HBeAg expression among WT, W3S and W3S (N146G), in either the medium or cytoplasm (indicated as cell), at either 5 or 7 days after transfection (Figure 2A,B). Furthermore, the expression of large S from the cells transfected with the respective plasmids was also confirmed by IFA with an anti-preS1 Ab (Figure 2C). Intracellular core-associated DNA and extracellular particle-associated HBV DNA were quantified using qPCR. The results showed that intracellular core-associated DNA had a tendency to be increased in W3S and W3S (N146G) compared with the WT, at either 5 or 7 days post-transfection (Figure 2D). Interestingly, the inverse was found for extracellular virion production, which was decreased in W3S and especially in W3S (N146G) at either 5 or 7 days post-transfection (Figure 2D). This phenomenon seemed to be a mirror image of the findings for intracellular core-associated HBV DNA, suggesting that accumulation should happen when secretion was disturbed (Figure 2D). To further confirm these results, we tried to detect the viral replicative DNA by Southern blot analysis at 7 days post-transfection. Accordingly, extracellular particle-associated HBV DNA was clearly detected for the WT but not the W3S or W3S (N146G) mutant (Figure 2E, right). In the case of intracellular particle-associated HBV DNA, the maturated HBV replicative intermediates as RC-DNA (RC) and/or double-stranded linear DNA (DSL) were more abundantly detected in W3S and W3S (N146G) (Figure 2E, left), suggesting that the maturation of HBV genome replication should progress with accumulation in the cell.

### 3.3. The Physicochemical Properties of the WT and W3S Dane Particles Are Basically Identical

In order to test the physicochemical properties of WT and W3S particles, CsCl density gradient ultracentrifugation of the virion and/or the HBs particles from the transfected cell culture medium was performed; twelve fractions were collected from the top; and the densities, the preS1 ELISA and the quantity of particle-associated HBV DNA are shown (Figure 3A,B). The results demonstrated that WT and W3S virions could be produced at a density of 1.20 to 1.25 g/mL. On the other hand, we could not show the density of glycosylation-deficient HBV (pHB-W3S [N146G]) from the medium of transfected cells because there was almost no virus secretion into the extracellular space (data not shown). Taken together, these results indicate that W3S virions were produced at a similar pattern in CsCl density profiles to that of the wild type, and glycosylation itself affected HBV virion secretion, as reported previously [24].

### 3.4. Effect of Tunicamycin on HBV Replication and Virion Production

The N-linked glycosylation of proteins might be reduced and/or inhibited by drugs such as tunicamycin [35,36,37]. Cytotoxicity of tunicamycin was not obvious at concentrations less than 1 µM (Figure 4B). Therefore, to further validate our results, we transfected the cells with pHB-WT and then treated them with tunicamycin. A schematic of the experimental design is depicted in Figure 4A. Western blot results demonstrated that the glycosylation isoform of large S was efficiently reduced/inhibited by tunicamycin treatment (Figure 4C). Then, the extracellular virion production in the medium and intracellular core-associated HBV DNA was quantified by qPCR at day 6 post-transfection. As expected, the extracellular virion production was significantly decreased with tunicamycin treatment compared to untreated cells (Figure 4D, left). In contrast, the intracellular core-associated HBV DNA was increased in tunicamycin-treated cells (Figure 4D, right). These results further suggested that an imbalance of the glycosylated and non-glycosylated isoforms of HBsAg would hamper the efficient virion release.

## 4. Discussion

Mutations in the hepatitis B virus genome have been reported worldwide, and these mutations have been affecting the diagnosis of hepatitis B, the effectiveness of hepatitis B vaccination, and occasionally patient resistance to antiviral drugs for hepatitis B [27,30,38,39,40]. Such mutations are known to influence the viral replication and pathophysiology, and thereby to be involved in occult HBV infection and persistence [9,12,41,42,43]. On the other hand, HBsAg is a component of HBV envelope proteins, whose antigenicity and immunogenicity should be affected by mutations in the HBsAg/RT overlapping region, since the major antigenic determinant, known as the “a” determinant, lies in this region [27]. However, it has remained to be clarified whether antigenic and/or diagnostic escape of HBsAg mutations affects the viral replication and cellular pathophysiology.

In this study, we found that several HBV pol RT domain mutations seen in the HBs ORF were associated with massive glycosylation of HBsAg even due to having only single-conserved glycosylation site (N146), though none of them were reported as drug-resistant mutants [9,12]. Massive glycosylation of HBsAg due to mutations in the HBs/RT overlapping region could be assembled in Dane particles, but drastically reduced the virion secretion. In contrast, the mutations did not affect the expression or secretion of HBeAg.

The single-conserved N-linked glycosylation site was reported to produce a nearly 1:1 ratio of glycosylated/non-glycosylated isoforms of HBV envelope proteins [23]. Hyper-glycosylation due to the creation of more than one N-linked glycosylation in the HBsAg ORF either by natural or artificial mutations affected the HBV assembly and the virion secretion efficiency [23,25,26]. Though only a single N-linked glycosylation site at amino acid position 146 was present in both the wild-type and W3S HBsAg, the level of the glycosylated isoform was much higher in W3S compared to wild-type HBsAg, and two amino acids mutations in the HBs/RT region of W3S at P120K and T123D were responsible for this phenomenon as well as escape from HBs ELISA. Therefore, it was suggested that the amino acid properties around the conserved N-linked glycosylation site (N146) should affect the glycosylation ratios of the HBsAg [27].

Accordingly, since the region overlapped with the HBV pol RT region, the mutations would affect HBV replication efficiency, and drug-resistant HBV might be generated simultaneously [38,44]. In this W3S mutant, eight mutations in the RT domain were found, some of which were associated with massive glycosylation and silent antigenicity, but none of these have been reported as drug-resistant mutants (Figure 1B). To prove it, we tested the effect of entecavir (ETV) on the replication of W3S and found no resistance (data not shown).

To test particle formation of WT and W3S, cesium chloride (CsCl) density-gradient ultracentrifugation was conducted [45,46]. The results confirmed that the patterns of particle formation were similar between the wild type and W3S, and thus massive glycosylation itself did not affect HBV particle formation (Figure 3A,B), though more sample required to show the profile of W3S particles and secretion of W3S HBsAg, i.e., the mature virion, was tremendously affected (Figure 2) and so was the non-glycosylated mutant W3S (N146G). This suggested that balanced or appropriate glycosylation should be very important for HBV virus secretion.

It was reported that the viral load in cells and accumulation of misfolded proteins due to mutations might cause endoplasmic reticulum stress (ER-stress), and HBV mediated ER-stress might be responsible for HCC formation [47]. A recent study demonstrated that additional N-linked glycosylation sites in the major hydrophilic region (MHR) of HBsAg should be associated with HCC [26]. We tested whether accumulation of S proteins of W3S and W3S (N146G) in the cell caused ER-stress, but we found that they did not (data not shown).

Previous studies showed that HBsAg could be hyperglycosylated by the artificial creation of multiple functional N-linked glycosylation acceptor sites, which significantly reduced the virion release without affecting the subviral protein synthesis and release [23]. On the other hand, Osthole, a Chinese herbal medicine, significantly decreased the non-glycosylated isoform of the HBsAg in transfected cells and thereby reduced the extracellular virion production [48]. Ito et al. demonstrated that removal of the N-linked glycosylation site completely blocked the virion release [24]. In our study, the massively glycosylated (W3S) and glycosylation-deficient (N146G) HBV replicated normally, but the virion seemed to be stacked in the cells. Such stacked virions or HBs protein in the cells might be associated with HBV pathogenesis, as described previously [47].

Together with these previous results, our present findings might indicate that glycosylated/non-glycosylated isoforms of HBsAg must be maintained in a balanced ratio (1:1) for efficient HBV release, which, if true, confirms that both isoforms play a functional role. An imbalanced ratio of the glycosylated/non-glycosylated isoforms of HBsAg due to mutations causes virions to become stuck in the cells, and may also affect host cellular signaling for progression of HCC and glycosylation of HBsAg might be a target of treatment.

## Figures and Tables

**Figure 1 viruses-13-02328-f001:**
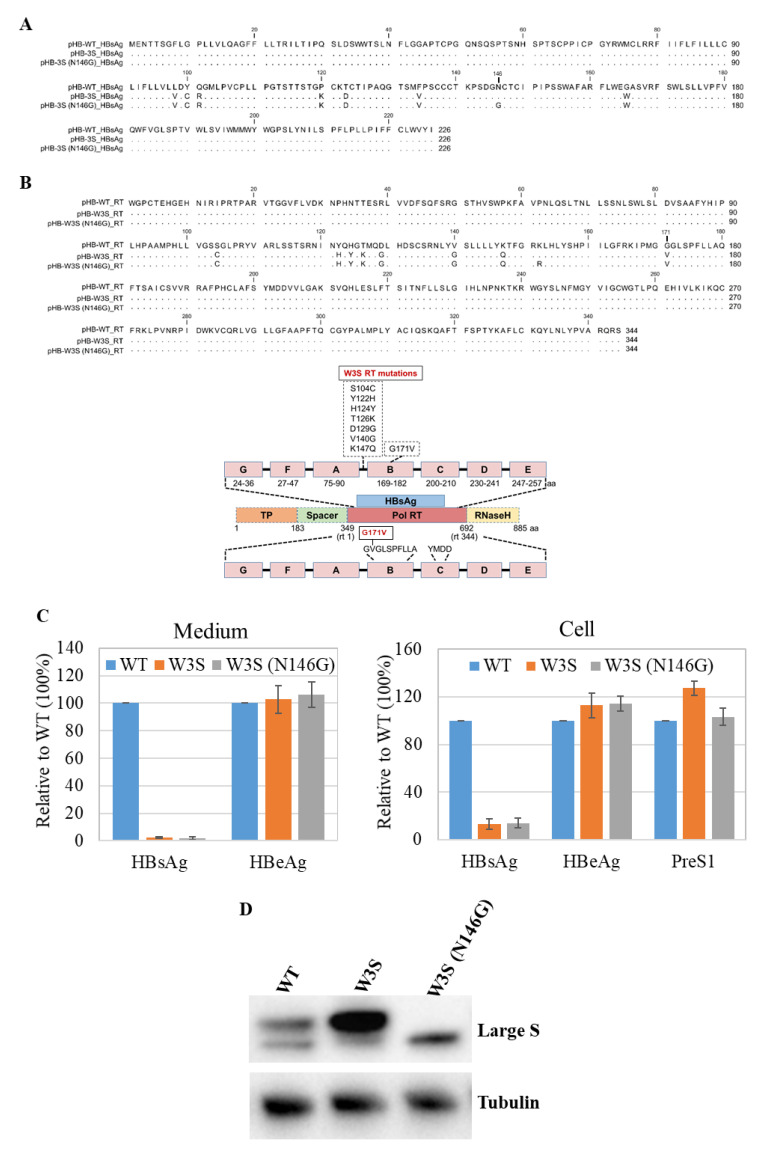
Amino acid mutations and antigen expression of HBV-WT, W3S, and W3S (N146G). (**A**) Alignment and comparison of the HBs amino acid sequences of WT, W3S, and W3S (N146G). (**B**) Upper: HBV pol RT amino acid sequence of WT, W3S, and W3S (N146G). Bottom: Relationship between HBs and HBV pol RT and mutations in the HBV-3S pol region. Identical amino acid residues are indicated by a dot. (**C**) Expression of HBsAg, HBeAg and preS1 (large S) in the soup and the cell. HBsAg, HBeAg, and preS1 ELISA were performed from 3 days post-transfection of HepG2 cells and/or medium. The results are presented as the average of three independent experiments with the standard deviation (SD). (**D**) Western blot analysis of cells at 3 days post-transfection using an antibody targeting the preS1 region.

**Figure 2 viruses-13-02328-f002:**
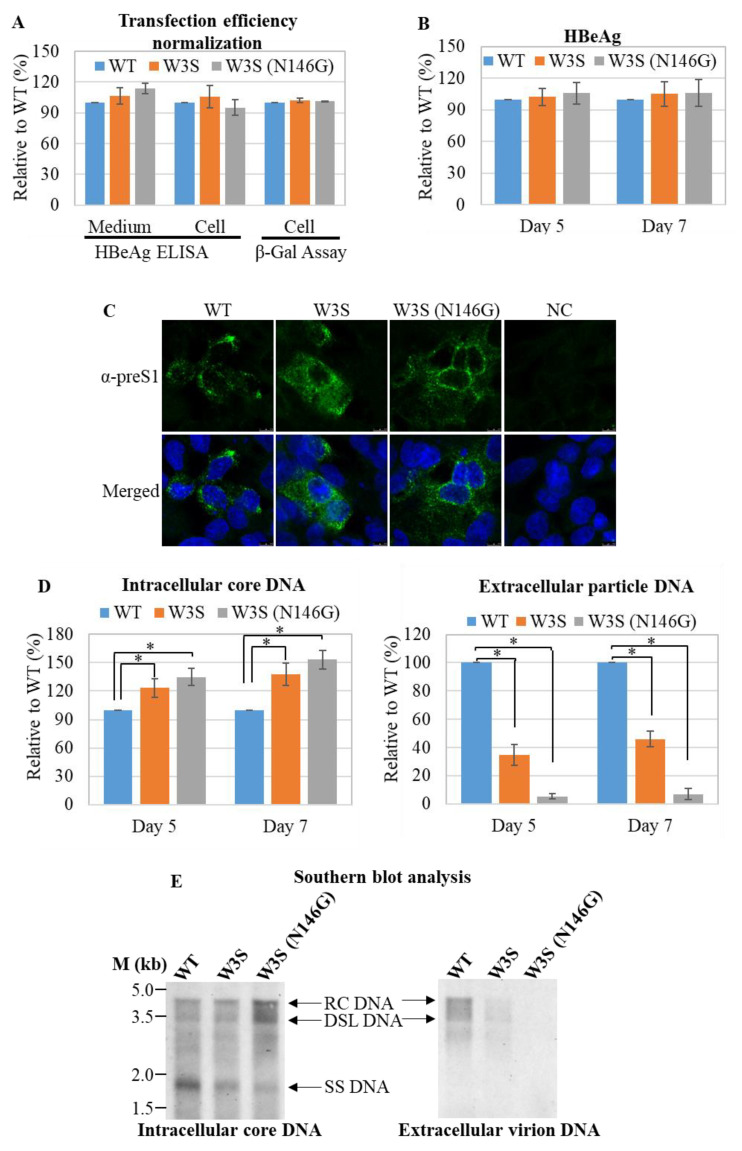
Replication and virion production efficiency of WT and mutant HBV. (**A**) Normalization of transfection efficiency by Beta-gal assay. The cells were transfected according to the protocol described in the Materials and Methods section. The cells were collected at 3 days post-transfection, lysed, and subjected to β-gal assay. HepG2 cells were transfected with wild type and mutant HBV replicating plasmids (pHB-WT, pHB-W3S or pHB-W3S [N146G]). Cells and medium were collected at days 3, 5, and 7 post-transfection and subjected to further analyses. (**B**) HBeAg ELISA analysis of the medium. (**C**) Immunofluorescence analysis of the transfected cells. Subcellular localization of preS1 (large S) protein was analyzed with an anti-preS1 antibody (green) and nucleus was counter-stained with DAPI. (**D**) Quantitative real-time PCR (qPCR) of extracellular virion-associated and intracellular core-associated DNA. * indicates statistical significance at *p* < 0.05. (**E**) Southern blot analysis.

**Figure 3 viruses-13-02328-f003:**
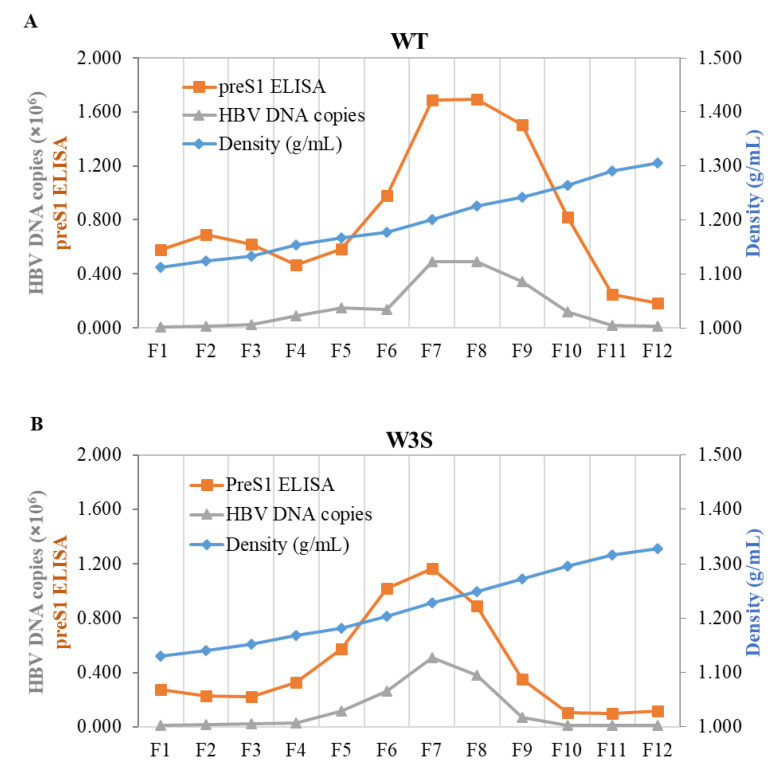
Analysis of the HBs and Dane particle formation pattern. HepG2 cells were transfected with the wild-type and mutant HBV-producing plasmids. The media were collected at 7 days post-transfection, PEG-precipitated, and subjected to CsCl density gradient centrifugation. Density, preS1 ELISA and qPCR analyses were carried out from a total of 12 collected fractions. (**A**) HBs and Dane particle formation pattern of wild-type HBV. (**B**) HBs and Dane particle formation of the massively glycosylated mutant HBV (W3S).

**Figure 4 viruses-13-02328-f004:**
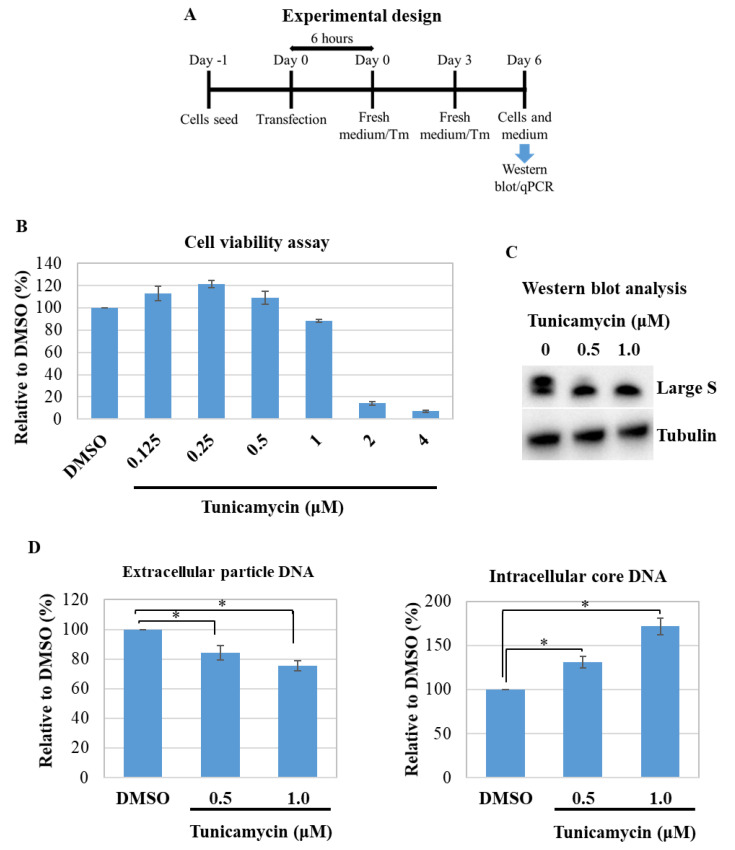
The effect of tunicamycin-induced removal of the glycosylation isoform on HBV replication and virion production. (**A**) Experimental design. The transfected HepG2 cells were treated with the indicated concentration of tunicamycin, and the cells and medium were collected for qPCR and Western blot analysis. (**B**) The cell viability assay. The cells were treated with tunicamycin at the indicated concentration at days 1 and 3 post-seeding, and a viability assay was performed at day 6. (**C**) Western blot analysis from the cell lysates. (**D**) Left: Quantitative real-time PCR (qPCR) of the extracellular virion production. Right: Intracellular core-associated DNA from the transfected and tunicamycin-treated cells. * indicates statistical significance at *p* < 0.05.

## Data Availability

Not applicable.

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
