# Peer review of "Analysis of the Physicochemical Properties, Replication and Pathophysiology of a Massively Glycosylated Hepatitis B Virus HBsAg Escape Mutant"

_viruses, 2021, doi:10.3390/v13112328_

Round 1
Reviewer 1 Report
Hossain et al. investigated the effects of a specific mutation, responsible for hyperglycosylation, in a diagnostic escape mutant (W3S) on HBeAg, HBsAg and HBV particle secretion. Furthermore, they characterized the particles by their specific CsCl densities and in a last step used the drug tunicamycin to block glycosylation in wtHBV.
While the overall the concept of glycosylation and especially hyperglycosylation in this study is interesting to field, the impact is not clear by showing this for the specific reported variant only.
By exchanging N146G Hossain et al. block glycosylation and show that both hyper- and no-gylcosylation lead to particle secretion deficiency. Most of these experiments for wtHBV, however, have already be done (Julithe et al. 10.1128/JVI.01161-14). In a last experiment they claim, that a specific drug, called tunicamycin, block glycosylation of PreS1 and thus lead to a block of HBV particle secretion in wtHBV. It would be very interesting if Hossain et al. are able to rescue HBV particle secretion by applying this drug to their hyperglycosylated variant and achieving a 1:1 ration again.
Minor:
The authors should mention that the Pol K152R mutation is due to the N146G mutation, which might not be obvious for some readers (259-261).
The authors should reorder the paragraphs. After analysis of HBeAg and HBsAg (Fig 1) the particle secretion (Fig. 3) should be shown, since this has a direct impact on the experiments with the particle properties (Fig. 2).
There are no indications in the figures which statistical analysis was used.
The authors should explain in the introduction how the hyperglycosylation is caused according to their former publication.
They authors might show the effect of glycosylation on particle formation and secretion using wtHBV with N146G to compare it with their mutant.
The authors claim that virions are produced at a similar level (293-294), however this does not match to their result in Fig. 3D.
The authors say, that there is comparable LS expression (268-269), however, Western Blot shows clearly more LS in the W3S variant compared to wtHBV or W3S-N146G in Fig. 1D.
The authors should quantify the overall HBV-DNA in the supernatant together with the cells to show that replication and viral titer of the mutants and wtHBV is comparable.
Major:
The authors claim that there is no difference in the HBeAg and PreS1 expression the medium and/or the cells, however in Fig. 1C there is no PreS1 data shown for medium (263-264).
It is not obvious, how the authors distinguish between HBV DNA from HBV replication and DNA from Plasmid transfection in Figure 3D. It is also not clear what the authors mean with “core-associated DNA”.
In the discussion the authors claim that they found several HBV pol RT domain mutations seen in the HBs 371 ORF were associated with massive glycosylation of HBsAg, however they only investigated N146G mutation.
The authors suggest that HBV Dane-Particle secretion is inhibited through the non- or hyperglycosylation by showing reduced amounts of HBV particle DNA in the supernatant. To proof this claim, the authors should show that capsid protein is accumulated in pHBV-W3S and pHBV-W3S transfected cells using western-blot or Immunofluorescence.
The authors nicely show a reduced glycosylation of wt-PreS1 using tunicamycin. Since they claim that for a stable particle formation and secretion a ration of 1:1 would be ideal, they could proof this by using tunicamycin in different concentrations on W3S transfected cells.
Author Response
Response to Reviewer 1 Comments
Point 1: Hossain et al. investigated the effects of a specific mutation, responsible for hyperglycosylation, in a diagnostic escape mutant (W3S) on HBeAg, HBsAg and HBV particle secretion. Furthermore, they characterized the particles by their specific CsCl densities and in a last step used the drug tunicamycin to block glycosylation in wtHBV.
While the overall the concept of glycosylation and especially hyperglycosylation in this study is interesting to field, the impact is not clear by showing this for the specific reported variant only.
Response 1: Thank you very much for reviewing our manuscript and for your appreciation and comments. Your excellent suggestions and comments helped us to improve the manuscript quality.
Point 2: By exchanging N146G Hossain et al. block glycosylation and show that both hyper- and no-gylcosylation lead to particle secretion deficiency. Most of these experiments for wtHBV, however, have already be done (Julithe et al. 10.1128/JVI.01161-14). In a last experiment they claim, that a specific drug, called tunicamycin, block glycosylation of PreS1 and thus lead to a block of HBV particle secretion in wtHBV. It would be very interesting if Hossain et al. are able to rescue HBV particle secretion by applying this drug to their hyperglycosylated variant and achieving a 1:1 ration again.
Response 2: Yes, Julithe et al. 2014 primarily focused on the wtHBV and checked the effect of glycosylation deficient and hyperglycosylated HBsAg containing 5 glycosylation sites (N-5). In our study, the W3S HBsAg showed extremely massive glycosylation due to the presence of a conserved single glycosylation site (146N). As the massively glycosylated HBsAg reduced the virus secretion efficiency. Therefore, here we checked whether any imbalance of 1:1 HBsAg ratio due to a single glycosylation site affecting the viral secretion efficiency by the partial suppression of the glycosylation in wtHBV, and the results are as expected.
Minor:
Point 3: The authors should mention that the Pol K152R mutation is due to the N146G mutation, which might not be obvious for some readers (259-261).
Response 3: Mentioned in the revised manuscript (Page: 6, Lines: 261-263).
Point 4: The authors should reorder the paragraphs. After analysis of HBeAg and HBsAg (Fig 1) the particle secretion (Fig. 3) should be shown, since this has a direct impact on the experiments with the particle properties (Fig. 2).
Response 4: The paragraph (sections) have been reordered (Pages: 8-11, Lines: 309-365).
Point 5: There are no indications in the figures which statistical analysis was used.
Response 5: Now mentioned in the revised manuscript (Page: 6, Line: 245).
Point 6: The authors should explain in the introduction how the hyperglycosylation is caused according to their former publication.
Response 6: Mentioned in the introduction (Lines: 93-96).
Point 7: They authors might show the effect of glycosylation on particle formation and secretion using wtHBV with N146G to compare it with their mutant.
Response 7: The effect of glycosylation on extracellular particle formation and secretion using wtHBV and N146G have been shown in figure 2 by qPCR and Southern blot analysis. However, due to the lack of secretion of Dane particle, properties of N146G virus in the CsCl density gradient centrifugation could not be possible.
Point 8: The authors claim that virions are produced at a similar level (293-294), however this does not match to their result in Fig. 3D.
Response 8: The sentence has been corrected (Page: 10, Lines: 357-358).
Point 9: The authors say, that there is comparable LS expression (268-269), however, Western Blot shows clearly more LS in the W3S variant compared to wtHBV or W3S-N146G in Fig. 1D.
Response 9: The glycosylated band of W3S variant is much stronger than the faint non-glycosylated one, whereas the glycosylated and non-glycosylated band of wtHBV is the almost same ratio.
Point 10: The authors should quantify the overall HBV-DNA in the supernatant together with the cells to show that replication and viral titer of the mutants and wtHBV is comparable.
Response 10: Figure 2D and E clearly showed differences and vice versa of intracellular core associated DNA quantification and extracellular Dane particle quantification from the supernatant. In the case of intracellular core associated DNA quantification, the complete virion (Dane particle) might not be formed whereas in the case of extracellular virion quantification from the supernatant complete virions (Dane particle) has been formed. Therefore, the core associated particle stuck in the cells might further affect the replication. Therefore, the overall HBV-DNA in the supernatant together with the cells may not exactly reflect the actual results.
Major:
Point 11: The authors claim that there is no difference in the HBeAg and PreS1 expression the medium and/or the cells, however in Fig. 1C there is no PreS1 data shown for medium (263-264).
Response 11: The information has been corrected in the revised manuscript (Page: 6, Lines: 265-268).
Point: 12: It is not obvious, how the authors distinguish between HBV DNA from HBV replication and DNA from Plasmid transfection in Figure 3D. It is also not clear what the authors mean with “core-associated DNA”.
Response 12: We described in the method section the protocol of the DNA extraction from cells and medium (Page: 5, Lines: 204-217). From the cells, only the encapsidated DNA was extracted excluding the replicating DNA, RNA, and transfected DNA. After breaking the cell membrane, the non-core particle-associated cytoplasmic DNA and RNA were degraded by treating with 2 μL (10 unit) DNase I (Takara-Clontech) and 5 μL (0.5 μg/ml) RNase A (Roche, Switzerland) in the presence of 5 mM MgCl2 and 5 mM CaCl2 for at least 3 h at 37°C. Then the HBV DNA within the core particle (Called core associated DNA) was extracted to quantification and Southern blot. This protocol was already previously approved (Okuyama-Dobashi et al 2015; Ref #32 in the manuscript).
Point 13: In the discussion the authors claim that they found several HBV pol RT domain mutations seen in the HBs 371 ORF were associated with massive glycosylation of HBsAg, however they only investigated N146G mutation.
Response 13: The sentence has been modified in the revised manuscript (Lines: 401-404). Though W3S is massively glycosylated it has only a single conserved glycosylation site at amino acid position 146 (N146). This study and our previous report showed that replacement of only N from the aa position 146 completely abolished the glycosylation. Therefore, other mutations could not facilitate glycosylation until the presence of N146. Therefore, we focused on the N146G.
Point 14: The authors suggest that HBV Dane-Particle secretion is inhibited through the non- or hyperglycosylation by showing reduced amounts of HBV particle DNA in the supernatant. To proof this claim, the authors should show that capsid protein is accumulated in pHBV-W3S and pHBV-W3S transfected cells using western-blot or Immunofluorescence.
Response 14: The accumulation of core associated DNA in the cells has been confirmed by qPCR and Southern blot analyses (Figure 2).
Point 15: The authors nicely show a reduced glycosylation of wt-PreS1 using tunicamycin. Since they claim that for a stable particle formation and secretion a ration of 1:1 would be ideal, they could proof this by using tunicamycin in different concentrations on W3S transfected cells.
Response 15: Tunicamycin reduced the glycosylation ratio of wtHBsAg. The W3S HBsAg massively glycosylated and tunicamycin may reduce the glycosylation ratio as well. On the other hand, tunicamycin upregulates the ER stress and there is another mechanism of ER-stress and HBV pathogenesis. Therefore, it is very sensitive and complex to maintain the exact glycosylation and non-glycosylation ratio by tunicamycin treatment.

Reviewer 2 Report
The article investigates the characteristics of HBV HBsAg WT and mutant that want to find the treatment at HCC and HBsAg glycosylation. There are some minor problems need to be solved.
- The title can be modified to fit the glycosylation of HBsAg.
- Figure 1(A). The site of N146G can be indicated or labeled for readers to read easily.
- Figure 2 and Figure 4 should be higher resolution.
Author Response
Response to Reviewer 2 Comments
The article investigates the characteristics of HBV HBsAg WT and mutant that want to find the treatment at HCC and HBsAg glycosylation. There are some minor problems need to be solved.
Thank you very much for reviewing our manuscript and for the appreciation and comments.
Point 1: The title can be modified to fit the glycosylation of HBsAg.
Response 1: The has been modified according to your suggestion (Page: 1, line: 3).
Point 2. Figure 1(A). The site of N146G can be indicated or labeled for readers to read easily.
Response 2: The site of N146G is indicated in the revised manuscript.
Point 3: Figure 2 and Figure 4 should be higher resolution.
Response 3: Figure 3 (original manuscript figure 2) and figure 4 have been replaced.
